# Effect of Sodium Hypochlorite Concentration on Electronic Apex Locator Reliability

**DOI:** 10.3390/ma15030863

**Published:** 2022-01-23

**Authors:** Franck Diemer, Emma Plews, Marie Georgelin-Gurgel, Lora Mishra, Hyeon-Cheol Kim

**Affiliations:** 1Department of Conservative Dentistry and Endodontics, CHU de Toulouse, Clement Ader Institute, 31400 Toulouse, France; franck.diemer@wanadoo.fr (F.D.); emma.plews@gmail.com (E.P.); marie.gurgel@icloud.com (M.G.-G.); 2Department of Conservative Dentistry and Endodontics, Institute of Dental Sciences, Siksha ‘O’ Anusandhan, Bhubaneswar 751003, Odisha, India; 3Department of Conservative Dentistry, School of Dentistry, Dental Research Institute, Pusan National University, Yangsan 50612, Korea

**Keywords:** electronic apex locators, working length, concentration, sodium hypochlorite

## Abstract

This ex vivo study aimed to measure the performance of an electronic apex locator (EAL) in the presence of sodium hypochlorite irrigants with different concentrations. Two EALs (Root ZX Mini and Locapex 6) were used to locate the apical foramen in 10 extracted single-rooted teeth in the presence of 0.5%, 2.5% and 5% sodium hypochlorite. Visual working lengths were also determined using #10 K-file under a microscope before the electronic measurements were made. The performance of both EALs was compared for the electronic working lengths determined under the different concentrations of sodium hypochlorite. A multiple-way ANOVA and PLSD Fisher’s test with an α risk fixed at 5% were conducted. There were no statistical differences in the working lengths determined by both EALs between the three groups with different concentrations of sodium hypochlorite and their visual control measurements. When a ± 0.5 mm margin was applied, the Root ZX Mini and the Locapex 6 presented 88% and 83% accuracy, respectively. Sodium hypochlorite concentration in irrigants does not affect the accuracy and reliability of either the Root ZX Mini or the Locapex 6. Electronic apex locators are reliable with any concentration of sodium hypochlorite irrigants.

## 1. Introduction

Error-free determination of working length (WL) is instrumental in ensuring the integrity of the periapical area during an endodontic procedure. The working length is defined as the distance between a coronal reference point and the apical point, consensually defined by the apical constriction, at which canal preparation and obturation should terminate [1,2]. Traditionally, the WL determination was performed using the radiographic image. However, the traditional radiographic imaging technique has inherent limitations, including distortion, magnification, interpretation variability and a lack of 3-dimensional representations [3]. It is a well-established fact that the radiographic apical vertex does not necessarily coincide anatomically with the minor foramen, major foramen and cementoenamel junction [4].

The shortcomings of the radiographic technique are easily overcome using an electronic apex locator (EAL). The EALs have been presented as valid instruments for identifying the apical constriction and determining working length as an alternative to the radiographic method [5]. The new generation of EALs has proven to equal or be more reliable devices in detecting WL compared to radiographs in clinical scenarios, and the correct use of a calibrated EAL may reduce the need for further radiographs [6]. 

Root ZX developed by Kobayashi and Suda [6] and manufactured by the Morita Co. (Kyoto, Japan) has been one of the most investigated EALs by the scientific community since its introduction. This device shows excellent performance, which makes it the gold standard EAL. It is a third-generation apex locator based on the “ratio method” that uses dual-frequency and comparative impedance principles [6,7].

However, the reliability of EALs has been questionable in the presence of irrigants in the root canal space. In particular, the presence of electroconductive solutions such as sodium hypochlorite (NaOCl) present inside the canal significantly reduced the impedance and, therefore, resulted in shorter measurements, whereas more extended measurements were detected in the lower electroconductive solution [8]. However, only a few published studies have evaluated the accuracy of Root ZX in the presence of multiple or increasing concentrations of NaOCl [8,9]. Furthermore, there is no research on the performance of Locapex 6 in the presence of electroconductive irrigating solutions. 

Therefore, this ex vivo study aimed to assess the impact of three different concentrations of NaOCl on the accuracy of two electronic apex locators: Locapex 6 (Ionyx, Bordeaux, France) and Root ZX Mini (Morita Co.). The null hypothesis was that there were no differences among the three different concentrations of NaOCl and between the EALs.

## 2. Materials and Methods

The protocol of this study using extracted teeth was approved by the institutional review board of Institute of Dental Sciences, Siksha ‘O’ Anusandhan (SOA/IDS/IRB 2021/9-I). Thirty single-rooted caries-free teeth, which were freshly extracted due to periodontal or orthodontic reasons with mature apices, were collected from the Department of Oral surgery. Twenty teeth from the collected teeth samples, which had either apical resorptions, metallic restorations, prosthetic or endodontic treatments, were excluded from the study. Ten teeth were finally selected. 

Incisal edge or cusps of the included teeth were flattened with a 12 mm cylindrical diamond burr (Komet, Paris, France) with a high-speed handpiece (NSK, Tochigi, Japan) to create a stable and reliable coronal reference point. The access cavity in each selected specimen was made with a 10 mm spherical diamond bur (Komet, Paris, France). An initial #8 K-file (Thomas, Bourges, France) was introduced to explore the canal and determine the initial visual-WL (VL) by advancing the file until the tip was tangential to the major foramen under a stereomicroscope (Wild M3B; Leica, Wetzlar, Germany) at 16× magnification. The rubber stop was then adjusted to the coronal reference point and the WL (VL) was measured. The value was recorded using an endodontic ruler and microscope magnification, with 0.1 mm precision. 

Two EALs were used in this study: Locapex 6 (LPX6) and Root ZX Mini (MRZX) (Table 1). The MRZX is the miniature version of the Root ZX (Morita Co.). Its functioning, accuracy and reliability were reported as comparable to that of Root ZX [10], however Locapex is a relatively new device introduced recently in the dental scenario. The manufacturer claims that LPX6 gives a 100% digital measurement without deviation in measurement accuracy at any time. It has a signal filtering system that eliminates false signals during the progress of the file in the canal (virtual apex) [11]. 

Tooth specimens and labial clips for each EAL were embedded in a plastic container filled with an unset mixture of alginate (Zelgan; Dentsply Sirona, Versailles, France). The alginate was mixed according to the manufacturer’s instruction and was left to set (Figure 1). 

The irrigating solutions were freshly prepared and divided into three groups according to the concentration of NaOCl: An original concentration of 9.6% NaOCl (Nectar, Nîmes, France) was diluted with sterile water for Irrigant 1 (IR1) with 0.5% NaOCl, Irrigant 2 (IR2) with 2.5% NaOCl and Irrigant 3 (IR3) with 5% NaOCl. 

The irrigation solutions were then used in order of lowest to highest concentration. When testing the different concentrations of NaOCl, the previously applied irrigation solution was aspirated, and canal was dried with paper point (Pierre Roland, Gustave Eiffel Merignac, France) to avoid concentration modifications. The root canal was flushed with 1 ml of irrigant, delivered using a side slotted and vented needle (27-gauge, Patterson Dental, Saint Paul, MN, USA). After irrigation, the pulp chamber was gently air-sprayed and coronal part of the canal was partially dried with optimal sized paper points. 

For each specimen tooth, the electronic WLs were measured in IR1, IR2 and IR3 recorded using both EALs, LPX6 and MRZX. Additional irrigation using 0.5 mL irrigants and drying process using air spray and paper points was repeated between each electronic WL measurement. The size #10 K-file connected to file holder was advanced till the “APEX” signal flashed on the screen for the LPX6 and the first red bar next to the “APEX” signal flashed in the MRZX. The rubber stop was adjusted to the coronal reference point, the file was taken out and the WL was confirmed using an endodontic ruler under a dental operating microscope (Leica M320; Leica Microsystems, Wetzlar, Germany) at 10× magnification. Two measurements per EAL were recorded for each tooth in each group. The same procedure was repeated for all groups IR1, IR2 and IR3. 

The statistical analysis was performed with StatView 5.0 (SAS Institute, Orange, CA, USA). VL and WL measured using LPX6 and MRZX were compared, regardless of NaOCl concentration, using a multiple-way ANOVA test and PLSD Fisher test to assess the comparability of both methods (visual and electronic). The Fisher test was then used to study the influence of NaOCl concentration on each EAL separately. For all tests, the α risk was set at 5%.

## 3. Results

The test subjects (root canals) were deemed comparable and formed a homogenous group. The PLSD Fisher test showed that there was no statistical difference between the VL and EAL measurements (*p* > 0.05). The WL measured visually (VL) and electronically (MRZX and LPX6) were not statistically different (ANOVA, *p* = 0.807).

In a direct comparison between the two EALs in the presence of various concentrations of NaOCl, both LPX6 and the MRZX showed no significant statistical difference when comparing the measured WL (Table 2).

MRZX and LPX6 were able to locate major foramen 88% and 83% of the time, respectively with a margin of error of ± 0.5 mm, regardless of NaOCl concentrations. When this margin is increased to ± 0.7 mm, the accuracy rates reached 97% for MRZX and 90% for LPX6 (Table 3).

When considering the individual concentrations of the NaOCl solution, LPX 6 appeared to be less accurate when the concentration increased. For the LPX 6, the precision rate dropped from 95% in group IR1 to 80% in IR2 and 75% with IR3 at or near 0.5 mm length from the apex (Table 2). On the other hand, MRZX was shown to be stable irrespective of the concentration of NaOCl, with a constant 90% precision rate in the presence of 2.5% and 5% NaOCl.

## 4. Discussion

An accurate WL is very important to achieve good clinical endodontic prognosis [2]. A short WL may lead to the infected tissue and bacterial organisms being left in the canal lumen, and an over or extended WL may produce a mechanical defect at or around the portal of the exit or periapical pathosis by inoculating the bacteria to the periapical tissue [12,13]. 

The development of EALs has enabled more accurate measurements of the apical constriction where the working length is terminated [14]. EALs use the human body to produce an electrical circuit [15]. They have two sides: one is connected to the endodontic file in the root canal, and the other is connected to the patient’s lip. The electrical circuit is completed when the tip of the dental file reaches the periodontal tissue at the apex. The electrical resistance of the EAL and the resistance between the file and the oral mucosa when they become equal results in a beeping noise, which indicates that the apex has been reached. There is a high correlation between the electronic and direct root canal length measurements [16].

The principle of operation consists in comparing the resistance of currents with two or more markedly different frequencies. During the shift of the file in the canal the resistance decreases to a larger extent for the current with high frequency, and the maximum difference is recorded at the place of connection of the pulp and the periodontal tissue, e.g., at the apical constriction. In the third-generation units the currents are analysed simultaneously and the differences or the ratio of the impedance values are calculated. In the next generation devices the currents are emitted and analysed separately. In the fifth generation apex locators the functional properties, such as the algorithm for calculating the properties of emitted currents, have been improved and a colour display to make the measurement easier to read has been added. Locapex 6 belongs to this 6th generation with an improved in-built algorithm.

The efficacy of EALs is of concern as they continuously work in wet canals. Irrigating solutions are vital to clean and completely disinfect the canal during endodontic treatment [17]. NaOCl is the main irrigant as it has a bactericidal effect and tissue resolving capability [14,15]. Meanwhile, the potency of NaOCl is the primary component which decides the efficiency of the chemical preparation. However, as it is an electroconductive solution, it may influence the electric resistance and the impedance during WL measurement using an EAL. Thus, this study evaluated the effect from the concentration of NaOCl.

This study concluded that NaOCl concentrations in irrigation solutions did not seem to adversely affect the reliability of EALs. These findings strongly indicate that EWLs can give a clinically accurate reading in the presence of NaOCl used in different concentrations, and any change in concentration does not affect the electrical results at such low currents. 

There are several in vitro studies that have evaluated the accuracy of MRZX in the presence of sodium hypochlorite irrigating solution and concluded that a dry or wet canal does not significantly affect the accuracy of MRZX [10,16,17,18,19]. These findings were supported in detail with the results of the present investigation. The accuracy rate of MRZX seemed to increase with the NaOCl concentration (IR1 85% and IR3 90%). This finding disagreed with other studies that concluded that in hypochlorite, the EAL measurements are significantly affected in new generation apex locators [20,21,22]. 

Another interesting finding from this investigation was that in the case of LPX6, the accuracy rate (±0.5 mm) proportionally decreased with NaOCl concentration (95%, 80% and 75% in group IR1, IR2 and IR3, respectively) (Table 2). This finding agreed with another article that reported the EWL measurements with LPX were more dispersed than other apex locators [23]. Venturi and Breschi [24], also found Root ZX to be more accurate in highly conducive conditions (5.25% NaOCl) than lower concentration and dry canals. The possible reasons for the difference in readings in dry or wet canals could be due to the difference in electrical conductivity of the solutions used [25].

Another explanation might be linked to the experimental model, which could affect the conductive properties of EALs. In a few studies, teeth were either immersed in 0.9% NaCl or a gelatine model [26]. At the same time, the alginate model provides an optimal and simulated environment close to the oral cavity, also offers stability and conductivity. It is also cost-efficient and practical [25]. Therefore, results in this investigation could be considered close to real-time results in the oral cavity. 

The ±0.5 mm margin of error is defined as being the strictest clinical tolerance and clinically acceptable [27]. The accuracy rate of MRZX at ± 0.5 mm increased with NaOCl concentration, going from 85% in IR1 to 90% in group IR2 and IR3 (Table 3). This finding was similar to another study that evaluated the performance of Root ZX in the presence of 2.125% and 5.25% NaOCl [28]. At ± 0.7 mm, the accuracy rate for LPX6 in IR3 increased by a considerable 15%, making it 90% accurate in 5.25% NaOCl (Table 3). Furthermore, all measurements (both EALs and all concentrations included) were within ± 1 mm, which would be the maximal tolerance interval. However, it can be inferred that measurements exceeding this 1 mm value may lead to incorrect WL estimations [2].

Similar to another study [12], the apical foramen was chosen as the apical reference point. It is a more reliable anatomical landmark and can be easily detected by the tip of a file under microscopic magnification, unlike the apical constriction, which is highly variable and sometimes non-existent [29].

An endodontic ruler is a routinely and most common WL transfer measuring device in dental practice [28], although in some studies they used a digital calliper or computer-based measuring system to measure the WL, which were considered to be more precise than the endodontic ruler [30]. However, all the measurements observed using an endodontic ruler were validated and recorded under a stereomicroscope with magnification. An acceptable method and precision up to 0.1 mm could be obtained [31,32]. 

Various ways to simulate in vivo conditions to determine the working length include 1% agar, gelatine, alginate, and flower sponge soaked in 0.9% saline and alginate models [33]. However, alginate was considered to be embedding media in several ex vivo endodontic apex locator studies [33,34,35,36,37]. In the present study, the model of choice was also alginate because it is acceptable and has been demonstrated to have good electroconductive properties [33,38]. Furthermore, the periodontal ligament was simulated more efficiently due to its colloidal consistency [39].

Under the limitations of this study, both MRZX and LPX6 were found to be reliable with 90% accuracy regardless of the concentration of NaOCl within an error margin of ±0.7 mm interval. Although this correlates with previous studies, there is the necessity for further research with additional parameters such as preflaring or EAL integrated endodontic motors used in NaOCl, heated NaOCl or other irrigating solutions.

## Figures and Tables

**Figure 1 materials-15-00863-f001:**
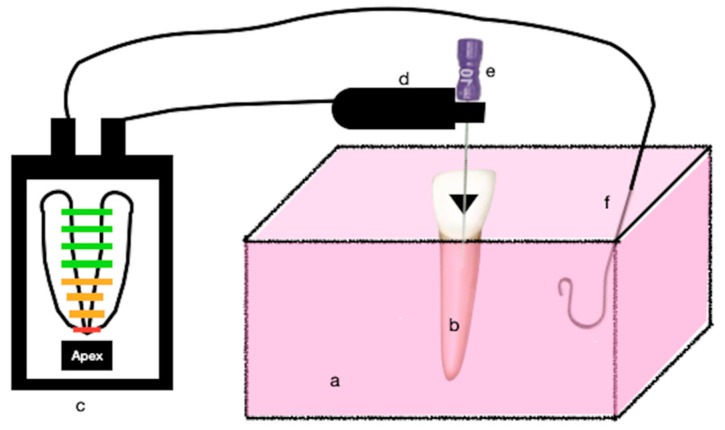
Schematic drawing of the test set-up for ex vivo alginate model for electronic working length measurement; (a) alginate, (b) tooth specimen, (c) apex locator, (d) file holder, (e) size #10-K-file and (f) lip clip.

**Table 1 materials-15-00863-t001:** Features of the electronic apex locators used in this study.

	Root ZX Mini	Locapex 6
Generation	4th	6th
Mechanism	- multi-frequency measurement system - frequencies are alternated rather than mixed - cancelling the need for signal filtering and eliminating the noise caused by non-ideal filters - RMS (Root Mean Square) level of the signal is measured, rather than its amplitude or phase	- adaptive type and combine the advantage of both 4th and 5th generation EAL - improved functional properties, such as the algorithm for calculating the properties of emitted currents - signal filtering system eliminates false signals during the progress of the file in the canal
Display	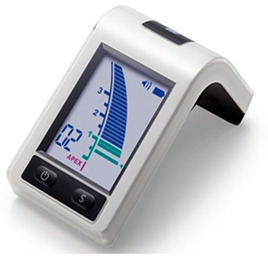	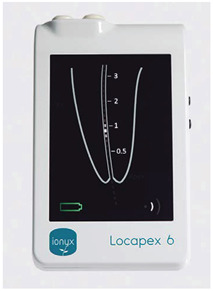

**Table 2 materials-15-00863-t002:** Comparison of the working length determined by Locapex 6 (LPX6) and RootZX Mini (MRZX) at different concentrations of NaOCl (PLSD Fisher *p*-value at different concentrations of NaOCl).

EAL\Groups	IR1–IR2	IR1–IR3	IR2–IR3	ANOVA
LPX6	>0.999	0.502	0.502	0.739
MRZX	0.689	0.745	0.939	0.912

There was no significant difference between groups (*p* > 0.05). EAL: electronic apex locator. IR1 (0.5% NaOCl), IR2 (2.5% NaOCl), IR3 (5% NaOCl).

**Table 3 materials-15-00863-t003:** Comparison of the accuracy between electronic apex locators and sodium hypochlorite concentration.

Electronic Apex Locator	RootZX Mini		Locapex 6	
Absolute Length Discrepancy	≤0.5 mm	≤0.7 mm	≤0.5 mm	≤0.7 mm
IR 1 (0.5% NaOCl)	17/20 (85%)	18/20 (90%)	19/20 (95%)	19/20 (95%)
IR 2 (2.5% NaOCl)	18/20 (90%)	20/20 (100%)	16/20 (80%)	17/20 (85%)
IR 3 (5% NaOCl)	18/20 (90%)	20/20 (100%)	15/20 (75%)	18/20 (90%)
All concentrations	53/60 (88%)	58/60 (97%)	50/60 (83%)	54/60 (90%)

## Data Availability

No new data were created or analyzed in this study. Data sharing is not applicable to this article.

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
