# Peer review of "Effect of Sodium Hypochlorite Concentration on Electronic Apex Locator Reliability"

_materials, 2022, doi:10.3390/ma15030863_

Round 1

Reviewer 1 Report

The paper "Effect of sodium hypochlorite concentration on electronic apex
locator reliability" is of some interest. The authors studied a situation that can have implications and clinical suggestions in daily practice. The study points out the performance of two different electronic apex locators in the presence of sodium hypochlorite irrigants with different concentrations.

Although it is quite similar to other studies already present in the literature, the study seems to me to be conducted correctly, with results close to those found in other researches.

The abstract is exhaustive, the introduction is clear enough; materials and methods are described correctly. The discussion, in my opinion, needs further more recent references that should strengthen and support the argoments presented by the authors. So in the references must be cited more recent articles.

Author Response

Authors responses: Thank you. we have added recent references (within 5 years from 2016) to the present investigation (Ref no 10,14, 16, 17, 21, 22, 33, 34, 35, 38). 

Reviewer 2 Report

The topic and the aims of the study are not original. The paper lacks novelty, the data presented do not add any knowledge to the already published literature and there is any update compared to the studies on the same argument. Moreover, the experimental design is poor and basic. Therefore, the manuscript is not acceptable for publication in Materials.

Author Response

Authors responses: We agree this critic but the new apex locators should be tested for these conditions in the study. We have added the necessity of the study further.

Reviewer 3 Report

I’ve reviewed the paper titled “Effect of sodium hypochlorite concentration on electronic apex locator reliability” with interest and following are my comments and suggestions.

  1. In the present study authors have presented the impact of different concentrations of NaOCl on the accuracy of two electronic apex locators but authors did not provide any justification and significance of this methodology.
  2. Is there any possibility that NaOCl irrigation would be a potential variable and influence the apex locator length determination?
  3. What is the role of different solutions and especially NaOCl on providing conducive conditions while recording working lengths?
  4. Authors are encouraged to describe the mechanism of electric current conduction in electronic apex locators used in this study.
  5. Additionally, it would be interesting to compare the results presented in this study with other methodologies for example using heated/non-heated solutions, use of EDTA, a combination of EDTA and NaOCl and the effect of canal preparation (K 10, 15, 30 etc).
  6. What is the role of “f. lip clip” shown in Figure 1, how the conductivity of alginate was determined?
  7. What do you think about the impact of integrated endodontic motors having apex locators with NaOCl and EDTA working length determination?

Author Response

I’ve reviewed the paper titled “Effect of sodium hypochlorite concentration on electronic apex locator reliability” with interest and following are my comments and suggestions.

  1. In the present study authors have presented the impact of different concentrations of NaOCl on the accuracy of two electronic apex locators but authors did not provide any justification and significance of this methodology.

Authors responses: There is no study yet which analyses the effect of Locapex 6 recently introduced apex locator. We have mentioned in introduction “The reliability of EAL has been questionable in the presence of irrigants in root canal space. Especially, the presence of electroconductive solutions like sodium hypochlorite (NaOCl) present inside the canal significantly reduced the impedance and, therefore, resulted in shorter measurements, whereas more extended measurements were detected in the lower electroconductive solution [8]. However, only a few published studies have evaluated the accuracy of Root ZX in the presence of multiple or increasing concentration of NaOCl [8, 9]”.

  1. Is there any possibility that NaOCl irrigation would be a potential variable and influence the apex locator length determination?

Authors responses: the presence of electroconductive solutions like sodium hypochlorite (NaOCl) present inside the canal significantly reduced the impedance and, therefore, resulted in shorter measurements, whereas more extended measurements were detected in the lower electroconductive solution. [Tinaz, A.C.; Sevimli, L.S.; Görgül, G.; Türköz, E.G. The effects of sodium hypochlorite concentrations on the accuracy of an apex locating device. J. Endod. 2002, 28, 160–162.]

  1. What is the role of different solutions and especially NaOCl on providing conducive conditions while recording working lengths?

Authors responses: Line 172-179: Irrigating solutions are vital to clean and completely disinfect the canal during endodontic treatment [17]. The efficacy of EALs is of concern as they continuously work in wet canals.

NaOCl is the main irrigants as it has bactericidal effect and tissue resolving capability [18,19]. As it is an electroconductive solutions, it may influence the electric resistance and the impedance during WL measurement using EAL[17]. Thus, this study evaluated the effect from the concentration of NaOCl.

  1. Authors are encouraged to describe the mechanism of electric current conduction in electronic apex locators used in this study.

Authors responses: Yes, Thank you, We have added (line:164-170)

  1. Additionally, it would be interesting to compare the results presented in this study with other methodologies for example using heated/non-heated solutions, use of EDTA, a combination of EDTA and NaOCl and the effect of canal preparation (K 10, 15, 30 etc).

Authors responses: Thank you. This is the first study on Locapex 6. Your suggestion would be highly valuable for a future strudy. We have described in the discussion section.

  1. What is the role of “f. lip clip” shown in Figure 1, how the conductivity of alginate was determined?

Authors responses: EAL is electronic device and for a completion of the current of the EPT is done with lip clip from the patient's lip.

Various ways to simulate in vivo conditions to determine working length include 1% agar, gelatin, alginate, and flower sponge soaked in 0.9% saline and alginate models [33]. However alginate is considered as embedding media in several ex vivo endodontic apex locator research [33–37]. In the present study the model of choice was also alginate because it is acceptable and has demonstrated to have good electroconductive properties [33,38]. Furthermore, the periodontal ligament was simulated more efficiently due to its colloidal consistency [39, 40].

  1. What do you think about the impact of integrated endodontic motors having apex locators with NaOCl and EDTA working length determination?

Authors responses: Both Apex locators are non-integrated apex locator, but this query would be good topic for a future study. The Apex locator in the integratd motor has the same principle for use and function, it might be similar. Thank you. 

Round 2

Reviewer 2 Report

I appreciated that the authors tried to improve the manuscript, but the manuscript is not original, and the experimental design is poor. Again, the manuscript is not suitable for an impacted journal, and it must be rejected.

Author Response

We agree with the critics. Authors believe the new devices need to be validated scientifically even the methodology is not new trial. Thank you. 

Reviewer 3 Report

Many thanks for providing the replies to most of my comments. However, I have a few suggestions which might help to improve the manuscript. 

  1. Please discuss and validate the claims of the manufacturer related to the performance of Locapex6 with the reference to most recent literature.
  2. The statement related to a single filtering system that eliminates false signals during the progress of the file in the canal requires further elaboration. 
  3. It would be interesting if you may consider providing a comparison in a table for both devices namely Locapex 6 (Ionyx, Bordeaux, France) and Root ZX Mini (Morita Co) in addition to the upgraded characteristic from the Locapex 5.
  4. There is a general agreement that the precision of an electronic working length measurement depends on the device used and the type of irrigation, and this study is targeting both of these points. Please provide your recommendations and justifications for the above two points. 
  5. Authors are encouraged to discuss differences reported in the literature between the methods in determining the EAL in the presence of sodium hypochlorite irrigants.

Author Response

1. Please discuss and validate the claims of the manufacturer related to the performance of Locapex6 with the reference to most recent literature.

Authors response: There is no single study depicting the performance of Locapex as it is recently introduced. That is why this study was conducted and important to understand how this particular EAL behaves in presence of NaOCl.

2. The statement related to a single filtering system that eliminates false signals during the progress of the file in the canal requires further elaboration. 

Authors response: We have added sentences about this issue in the M&M and discussion sections.

The purpose of this study was to see the effect from different concentration of the NaOCl. There is no specific point related to the false responses in the protocol of this study. We may try another study regarding false responses, if required.

3. It would be interesting if you may consider providing a comparison in a table for both devices namely Locapex 6 (Ionyx, Bordeaux, France) and Root ZX Mini (Morita Co) in addition to the upgraded characteristic from the Locapex 5.

Authors response: Explained in the new Table 1 and added in discussion.

The principle of operation consists in comparing the resistance of currents with two or more markedly different frequencies. During the shift of the file in the canal the resistance decreases to a larger extent for the current with high frequency, and the maximum difference is recorded at the place of connection of the pulp and the periodontal tissue, e.g. at the apical constriction. In the third-generation units the currents are analysed simultaneously and the differences or the ratio of the impedance values are calculated. In the next generation devices the currents are emitted and analysed separately. In the fifth generation apex locators the functional properties, such as the algorithm for calculating the properties of emitted currents, have been improved and a colour display to make the measurement easier to read has been added. Locapex 6 belongs to this 6th generation with improved in built algorithm.

4. There is a general agreement that the precision of an electronic working length measurement depends on the device used and the type of irrigation, and this study is targeting both of these points. Please provide your recommendations and justifications for the above two points. 

Authors response: Thank you. We gave reference from the following SR and meta analysis:

The precision of electronic WL measurement depends on the device used and the type of irrigation and is not influenced by the status of the pulp tissue (vital or necrotic). (Ref no 12)

5. Authors are encouraged to discuss differences reported in the literature between the methods in determining the EAL in the presence of sodium hypochlorite irrigants.

Authors response: Thank you. We have described it. There are several in-vitro studies which evaluated the accuracy of MRZX in presence of sodium hypochlorite irrigating solution and concluded that dry or wet canal does not significantly affect the accuracy of MRZX [10, 16–19].